# Ehrlichiosis in Dogs: A Comprehensive Review about the Pathogen and Its Vectors with Emphasis on South and East Asian Countries

**DOI:** 10.3390/vetsci10010021

**Published:** 2022-12-29

**Authors:** Muhammad Umair Aziz, Sabir Hussain, Baolin Song, Hammad Nayyar Ghauri, Jehan Zeb, Olivier Andre Sparagano

**Affiliations:** 1Department of Infectious Diseases and Public Health, Jockey Club College of Veterinary Medicine and Life Sciences, City University of Hong Kong, Kowloon, Hong Kong SAR 999077, China; 2Department of Veterinary medicine, University of Veterinary and Animal Sciences, Lahore 54000, Pakistan

**Keywords:** *Ehrlichia canis*, *Ehrlichia chaffeensis*, *Ehrlichia ewingii*, dog, epidemiology, tick borne diseases, therapeutic agents

## Abstract

**Simple Summary:**

In dogs, ehrlichiosis is caused by three Ehrlichial species, namely *Ehrlichia canis*, *E. ewingii*, and *E. chaffeensis*; however, *E. canis* is the pathogen that most affects platelets, monocytes, and granulocytes. Globally, *Rhipicephalus sanguineus* is mainly responsible for vectoring the Ehrlichia species; however, *Haemaphysalis longicornis* is also involved in vectoring this species in east Asian countries. This disease causes acute, sub-clinical, and chronic clinical complications. There is no preferable age or sex for ehrlichiosis. The disease can be diagnosed by various methods including microscopy, indirect immunofluorescence test (IFAT), and polymerase chain reaction (PCR). The treatment of choice for ehrlichiosis includes doxycycline, rifampicin, and minocycline. Overall, this review describes the infection rate of Ehrlichia in dogs, the associated reported prevalence in east and south Asian countries, currently used therapy, and associated vectors responsible for the disease transmission as well as future perspectives.

**Abstract:**

Ehrlichiosis in dogs is an emerging vector borne rickettsial zoonotic disease of worldwide distribution. In general, three Ehrlichial species (*Ehrlichia canis*, *E*. *ewingii,* and *E*. *chaffeensis*) are involved in infecting dogs. Among them, *E. canis* is the well-known etiological pathogen affecting platelets, monocytes, and granulocytes. Dogs act as a reservoir, while the main vector responsible for disease transmission is *Rhipicephalus sanguineus*. However, in east Asian countries, *Haemaphysalis longicornis* is considered the principal vector for disease transmission. This disease affects multiple organs and systems and has three clinical manifestations, including acute, subclinical, and chronic. Definitive diagnosis involves visualization of morulae on cytology, detection of antibodies through an indirect immunofluorescence test (IFAT), and DNA amplification by polymerase chain reaction (PCR). In canine ehrlichiosis, no predilection of age or sex is observed; however, Siberian Huskies and German Shepherds are more likely to develop severe clinical manifestations. Doxycycline, rifampicin, and minocycline are proven to be effective drugs against canine ehrlichiosis. This review is intended to describe a brief overview of Ehrlichia infection in dogs, its reported prevalence in east and south Asian countries, and the latest knowledge regarding chemotherapy and associated vectors responsible for the disease transmission. This manuscript also identifies the prevailing knowledge gaps which merit further attention by the scientific community.

## 1. Introduction

Ehrlichiosis is induced by a group of emerging rickettsial tick-borne pathogens of public importance that are Gram-negative obligate intracellular bacteria of the genus *Ehrlichia*, family Anaplasmataceae [1]. Ehrlichiosis in dogs is a significant vector-borne bacterial ailment spreading worldwide. It is also recognized as canine rickettsiosis, canine typhus, canine hemorrhagic fever, tropical canine pancytopenia, and tracker dog disease [2]. Numerous species of *Ehrlichia* are famous for infecting a wide range of animals. Among them, *E. canis* is the well-known etiological pathogen of canine ehrlichiosis affecting platelets, monocytes, and granulocytes [3,4]. Although *E*. *chaffeensis* is considered the main etiological agent of human monocytic ehrlichiosis (HME), it is also reported in canines [5]. A minimum of five different species of ticks (*Amblyomma americanum*, *Haemaphysalis longicornis, Rhipicephalus sanguineus*, *Haemaphysalis yeni,* and *Dermacentor variabilis*) have been identified as vectors transmitting clinical ehrlichiosis in dogs [6]. These disease-transmitting vectors become more potent during summer and spring seasons [7]. *Rhipicephalus sanguineus* (the brown tick of dogs) is generally considered the main vector responsible for canine ehrlichiosis [8,9]. In recent years, ehrlichiosis has been expanded to new regions which were thought to be disease free, such as northern China, temperate regions of the Indian sub-continent, and central and northern states of the USA [10].

Ticks are considered the second most prevalent hematophagous parasites after mosquitoes. Along with causing anemia, they also act as vectors for the transmission of many protozoan, bacterial, and viral diseases [11,12]. In recent years, ecological variations due to global warming, exponential increase in human population, deforestation, and frequent transport of pet animals from one continent to another have modified and enhanced the transmission patterns of all vector-borne pathogens around the globe [13]. In south and east Asia, regardless of the suitable climatic conditions for vectors and parasites and huge population of stray and pet dogs, scarce knowledge is available related to diagnosis, epidemiology, prevention, and control strategies associated with ehrlichiosis in dogs. However, with the expansion of a region’s economy and inculcation of foreign culture, the responsibilities of veterinarians have increased to devise control strategies about canine tick-borne ailments [14]. In less developed areas of east and south Asia, almost 75 percent of dogs are categorized as stray dogs, which further increases the risk of emergence of new tick-borne parasitic zoonoses [15].

Canine ehrlichiosis gained worldwide importance in the 1970s when a huge population of military German Shepherd dogs died during the Vietnam war [16]. *Ehrlichia canis* attacks cells of the immune system, particularly macrophages, leukocytes, and monocytes, developing a cytoplasmic membrane-bound cluster of bacteria termed morulae [17]. In the past, *E.canis* was not considered a zoonotic pathogen but recent studies have suggested its zoonotic significance in humans [18,19]. This disease affects multiple organs and systems and has three clinical presentations, including acute, subclinical, and chronic [20]. In the acute form, the main clinical symptoms in dogs are high fever, anorexia, lethargy, lymphadenomegaly, depression, epistaxis, splenomegaly, and petechial and ecchymotic skin hemorrhages. Ophthalmic lesions are also common and comprise chorioretinitis, papilledema, anterior uveitis, retinal hemorrhage, and occurrence of infiltrates at the retinal perivascular space [21]. The chronic form is more dangerous and characterized by anemia, paralysis, significant weakness, and death [20]. Due to several overlapping and non-specific clinical signs, diagnosis of the disease becomes challenging, and the need for alternative cutting edge molecular techniques is increasing [22]. This review is intended to describe a brief overview of *Ehrlichia* infection in dogs, its reported prevalence in east and south Asian countries, and the latest knowledge regarding chemotherapy and associated vectors responsible for the disease transmission. This manuscript also identifies the prevailing knowledge gaps which require the further attention of scientists.

## 2. Epidemiology

Ehrlichiosis is a disease of global importance, but it is more prevalent in sub-tropical and tropical regions. However, due to the chronic nature of the disease in some cases, accurate geographical distribution cannot be determined. The reason may be due to the fact that the clinical signs appear years after the first inoculation of the pathogen from ticks and after the canine species has traveled to non-endemic countries where this specific ailment might not be included in differential diagnostic lists by clinicians and scientists [23]. On the basis of different diagnostic methods, the prevalence in the south and east Asia ranges from 0.0% (South Korea) to 86.9% (India) [24,25]. The prevalence percentage of all *Ehrlichia* spp. related to canines in different countries of south and east Asia is presented in Table 1. Figure 1 depicts the prevalence of Ehrlichiosis across different regions.

In general, three *Ehrlichia* spp. (*E. canis, E. ewingii,* and *E. chaffeensis*) are involved in infecting dogs [66]. However, in Japan, variations among tick and *Ehrlichia spp.* are more evident. Instead of *R*. *sanguineus*, *Haemaphysalis longicornis* is the most commonly found tick species responsible for disease spread [67]. In recent times, a novel *Ehrlichia* species was found in *Ixodes ovatus* ticks, which showed phylogenetically close relationship with *E. chaffeensis* [68]. Similarly, *Rhipicephalus sanguineus* is rarely found in Korea. Instead, the most common tick species found in Korean dogs are *H. longicornis* and *Haemaphysalis flava* [5]. *H. longicornis* also possesses zoonotic significance because it has been found in close association with animals and humans [57]. Although *Amblyomma americanum* is the chief vector involved in *E. chaffeensis* infection, *Haemaphysalis yeni*, *testudinarium*, *Ixodes ricinus,* and *H. flava* have also been isolated as reservoirs [69,70,71].

Since its discovery, human monocytotropic ehrlichiosis (HME) has grown to be the most common life-threatening tick-borne disease. As the number of animal reservoirs and tick vectors have increased and people have increasingly settled in areas with high reservoir and tick populations, ehrlichiosis is being diagnosed as the primary cause of human diseases. The causative agent of HME, *Ehrlichia chaffeensis*, is a developing zoonosis that can have a variety of clinical presentations, from a mild febrile sickness to a fulminant disease marked by multiorgan system failure. Headache, fever, leukopenia, thrombocytopenia, and increased liver enzymes are among the clinical symptoms of human monocytic ehrlichiosis (HME). The majority of patients need medical assistance during the first 4 days of sickness, with symptoms often appearing 9 days on average after a tick bite. The majority of HME reports involve neurological symptoms [72].

## 3. Risk Factors Associated with Canine Ehrlichiosis

All dog breeds are susceptible to canine monocytic ehrlichiosis (CME). Nevertheless, because Siberian Huskies and German Shepherds are more prone to exhibit severe clinical symptoms, infection in these breeds frequently has the worst prognosis [72,73]. This was demonstrated experimentally by infecting German Shepherd and Beagle dogs with *E. canis* and seeing that the intensity of the cell-mediated immune response in the German Shepherd breed was lower than that of the Beagle breed dogs [72,73,74]. It seems that there is no predisposition of sex in disease occurrence. Although some published literature has detected increased seroreactivity in males, this can be explained by a higher chance of contact with tick species than females because of behavioral features [75]. There is no strong evidence of high disease prevalence in older dogs as well. However, some epidemiological studies show that seropositivity rates were higher in older dogs [50,76]. These results may be due to the higher likelihood of exposure to the *Ehrlichia* pathogen as the dog becomes older. Dogs living outdoors are more prone to have ehrlichiosis compared with pet dogs living indoors. Moreover, dogs living in non-sanitized enclosures with tick infestations are at risk of getting *E. canis* infection. Regular use of ectoparasiticidal drugs also decrease the risk of ehrlichiosis in dogs [77]. Moreover, environmental factors also play a major role in disease prevalence because high temperature and low humidity favors the growth of vectors, so animals living in those circumstances are at greater risk [78].Ticks spreading the disease are mentioning in Table 2.

## 4. Transmission Cycle

Brown dog tick (*Rhipicephalus sanguineus)* carries the pathogen from infected dog through blood meal during the acute phase of disease. After the blood meal, *E. canis* resides in the salivary glands and midgut of the carrier tick and it then spreads the pathogen to another healthy dog via its salivary glands during subsequent feeding [81]. Transstadial transmission is well-established for this pathogen in which the larval stage of tick becomes infected with *E. canis,* which can pass the bacteria to the next two stages (nymph and adult) and spread the pathogen during blood meals [82,83]. It has been observed that brown dog tick starts transferring the rickettsial pathogens within three hours of its attachment to a host [83]. If the *Rhipicephalus sanguineus* is transferred to a cold or temperate climate, due to the shifting of hosts, it can still remain active under such man-made protected kennel environments [84,85]. Moreover, enclosures of wild animals, abandoned houses, and kennel environments provide a perfect atmosphere for its reproduction. Under these suitable environmental conditions, only a single female tick was enough to infect and reproduce many subadults [86]. During the chronic or subclinical phase of the disease, the dog seems healthy but still acts as a carrier for this rickettsial pathogen. The only tick that becomes engorged during the acute phase can infect another healthy host. Moreover, this tick can spread the bacteria even after 155 days of its detachment [87]. Many studies have suggested that transovarial spread of the pathogen also occurs in Ixodid ticks and they maintain the bacteria through many generations in nature [88,89]. However, in a recent study, no proof of transovarial spread was found [90].

## 5. Pathogenicity

Unlike many Gram-negative bacterial pathogens, peptidoglycan and lipopolysaccharide are absent in the cell wall of this bacterium which may help the bacteria in resisting the host’s immune response. The cell wall of *E. canis* becomes very flexible due to the absence of these two materials, which in turn facilitates the pathogen in avoiding antibody attack from its host immune system. Other characteristic feature of this rickettsial organism is the lack of complex inner structures, which permits the production of sugars. The main energy source of this bacterium are amino acids [91]. The incubation period of *Ehrlichia* ranges from 8–20 days. This period is sequentially followed by subclinical, acute, or in some cases chronic form.

Pilli are absent in *Ehrlichia* so the outer membrane of this infectious agent helps in the attachment with the host cell. Once the pathogen enters the host cell and starts infection, it forms membrane-bound partitions (endosomes) and maintains its distinctive cytoplasmic shape. The main target of *E. canis* are mononuclear phagocytic cells. Monocytes are the most common cells to be infected both in canine and human hosts. In addition, *Ehrlichia* also attacks the other immune cells such as metamyelocytes, lymphocytes, and promyelocytes. In general, it is assumed that inside the cells only mononuclear phagocytic cells are able to uphold the productive pathogen [92]. On average, a single infected monocyte contains one to two morulae. The endosomal membrane formed by *Ehrlichia* protects the pathogen from the host and it multiplies within this apartment. The exact mechanism of their survival is still unclear but consequently, the pathogen may survive by modulating the host defense system [93]. In one study, researchers identified two paralogous proteins responsible for immune evasion, which may be due to the presence of poly (G-C) tracts in one of the proteins, suggesting that they have a role in facilitating chronic persistent infections and can help in phase deviation [91].

After infecting the monocytes, *E. canis* spreads to the whole lymphatic system including the liver and spleen, where it triggers the abnormal fast growth of cells and the increased size of these organs, described as hyperplasia. Further cell division and replication leads to bacteremia and eventually results in hemolysis. At this stage, severe clinical manifestations, such as high fever, anemia, and thrombocytopenia, can be observed [50]. Dogs suffering from persistent infection develop a more lethal form of chronic disease where the pathogen attacks the bone marrow and destroys the immune system. As a result, other opportunistic infectious agents further aggravate the situation. Severe thrombocytopenia leads to massive hemorrhages and death [91].

## 6. Clinical Signs

Clinical presentation due to ehrlichial infection can be varied and depends on many factors, such as the status of the immune system of the dog, virulence of the strain, and existence of co-infections with other tick/flea-borne diseases. Among all other members of the Anaplasmataceae family, *E. canis* appears to cause more intense clinical abnormalities [94,95,96]. In dogs, three ehrlichial species, namely *E. canis, E. ewingii,* and *E. chaffeensis,* can cause clinical disease [1,3]. The principal host cell targets for *E. canis* and *E. chaffeensis* are agranulocytes, while *E. ewingii* mainly targets the granulocytic white blood cells [2]. These pathogens can induce both clinical and subclinical complications. Clinical signs induced by ehrlichial species are often non-specific and overlapping. The disease can be acute or mild; however, in many cases, the animal becomes a carrier for an extended period of time without presenting any clinical manifestations. Typically, the incubation period for all ehrlichial species ranges from one to three weeks, and results in three possible disease presentations which may be categorized as acute, chronic, and subclinical [97]. The acute phase may last for 2 to 4 weeks and if the animal survives, the signs vanish even without chemotherapeutic treatment. However, some dogs become subclinical carriers after improvement and may become an important source of infection for months and years. In this phase, the animal apparently looks normal and healthy and does not present any clinically visible signs but upon hematological testing, mild thrombocytopenia can be detected [98]. Not all but some subclinically infected dogs may proceed to the chronic stage, which is the most fatal form of the disease, and which cannot be differentiated from the acute phase in clinical settings because most of the clinical manifestations are non-specific. The chronic form is also known as the myelosuppressive form in which it is difficult to distinguish it from acute bone marrow aspiration and complete blood count tests are necessary. Alternatively, hypoplasia of bone marrow and severe pancytopenia will confirm the presence of the chronic phase [99]. The possible factors that cause some dogs to enter the chronic phase are still unclear.

In naturally infected dogs, the common clinical findings are fever, pale mucosa due to anemia, lymphadenomegaly, bleeding disorders, hepatomegaly, lethargy, petechial and ecchymotic hemorrhages, vasculitis, and extended bleeding period during estrus [100,101,102]. Other less common signs of ehrlichiosis have also been defined and include diarrhea, exercise intolerance, neonatal death or abortion, vomiting, and mucopurulent or serous nasal and ocular discharge. Some old studies have mentioned polyarthritis and lameness as a sign of canine ehrlichiosis [103], but it is believed that this manifestation only appears in cases of co-infections. On physical examination, you may observe tick infestation particularly during the acute phase. In addition, other signs like ataxia, vestibular dysfunction and seizures, and chronic or myelosuppressive form also reveal stomatitis, scrotal or hind limb edema, jaundice, glossitis, and pyoderma [104]. Bleeding tendencies are also more frequent and severe in the chronic form of CME [102].

## 7. Clinical Pathology

Hematological abnormalities are variable and overlapping. However, severe drop in platelet count or thrombocytopenia is the principal abnormality observed in canine ehrlichiosis. This hematological finding is consistent in almost 80% of the animals, irrespective of the stage of the ailment. However, normal platelet count may not be the only reason to rule out ehrlichiosis [105,106]. In a retrospective study, decreases in total red blood cell count, pack cell volume (PCV), and platelet count while noticeable increases in basophil count were observed. Moreover, blood urea nitrogen (BUN) and creatinine levels were more elevated than normal [107]. Anemia is mostly non-regenerative along with lymphopenia, monocytosis, thrombocytopathy, hyperproteinemia, hypergammaglobulinemia, hypoalbuminemia, and hyperglobulinemia, which are some additional irregularities [108,109,110]. Values of neutrophils are inconsistent and both neutrophilia and neutropenia have been detected based on the phase of the severity. In the chronic form, aplastic pancytopenia, granular lymphocytosis, mild elevation in liver enzymes, and renal azotemia were found. In regions where this disease is endemic, CME should be the top differential in dogs having persistent lymphocytosis [111,112].

Histopathological and gross abnormalities in experimentally infected dogs include edema of the subcutaneous layer, ascites, anemia, jaundice, and emaciation. Cuffing of the lymphatic fluid in the cerebellum and brain is occasionally seen. Lungs of infected animals display vasculitis and interstitial pneumonia. Additionally, a flabby heart and whole heart dilatation can also be found. Grossly, the most frequent signs include apparent splenomegaly, multifocal lymph node necrosis, and widespread lymadenopathy [113].

## 8. Diagnosis

### 8.1. Microcopy

In blood smears, detection of ehrlichial organisms in the form of morulae is very rare. It is noticeable only in 4–6% of cases. The higher sensitivity of this method can be achieved by using a buffy coat smear [114,115]. Maximum detection percentage (50%) of morulae can be achieved by an expert cytologist that observes many microscopic fields using fluid from lymph nodes [116]. In the acute form of canine ehrlichiosis, the presence of *E. canis* morulae in mononuclear leukocytes using buffy coat, spleen, cerebrospinal fluid, and bone marrow provides a definitive diagnosis [117,118]. In one study, they measured the percentage of morulae detection in naturally infected dogs using lymph nodes, buffy coat, and their combination and found a 61%, 66%, and 74% detection rate, respectively [119]. Cytology is considered a laborious procedure having low diagnostic sensitivity but in acute cases it can provide earlier diagnosis even before serology. This method can also be helpful in documenting mix infections [92]. Bone marrow cytology is also used to differentiate myelosuppressive and non-myelosuppressive canine ehrlichiosis as well [12]. One of the limitations of microscopy is that it is extremely insensitive in the chronic and subclinical phases and unable to differentiate ehrlichial species. Moreover, it requires a lot of expertise to differentiate between other extraneous tissue structures and morulae [22].

### 8.2. Serology

Immunofluorescence antibody test (IFAT) and enzyme-linked immunosorbent assay (ELISA) can both be used for the diagnosis of ehrlichiosis [120,121]. Specific apparatus and technical manpower are required to run these tests. One of the benefits of using serological tests for the detection of infectious pathogens is that they can quantify the antibody titer and variations over time. Consequently, it gives an idea about the stage and intensity of infection. Quantitative serological methods have a high sensitivity and specificity rate as compared with rapid diagnostic tests [2]. The perseverance of moving antibodies can also have an advantage especially during the chronic phase of canine monocytic ehrlichiosis (CME) when the quantity of active pathogen in blood is too low to be measured through PCR or the pathogen is confiscated in tissues not normally submitted to run PCR [100,122]. Positive ELISA or IFAT only denote a past or present infection and does not indicate the current disease condition. You can get a positive result on the basis of antibody titer even when the infection is resolved in the past because antibodies may persist in the body for several months or even years [123]. Regardless of carrying an active infection, an animal can be serologically negative especially during the early stages of the ailment or during the incubation period. In ehrlichiosis, normally antibody synthesis does not start before 12 to 14 days after infection [124]. We propose that doubted cases must be accessed based on two to three serological tests performed within two to three weeks. Through this approach, we can check the trend of antibody titer (increasing, decreasing, or constant) and guess the present status of the malady. In a study, they mentioned the 4-fold growth of IgG as proof of a current infection [120]. Generally, it is assumed that there is no cross-reaction present between *Ehrlichia* and other Anaplasmataceae members. However, a possible such reaction has been defined between *A. phagocytophilum* and *E. canis*, mainly when the quantity of one pathogen was very high [125].

### 8.3. Molecular

PCR is very valuable in identifying these canine infectious ailments for many reasons. First, it can detect a minute amount of DNA with high precision. Secondly, instead of indicating past infections, it provides strong and clear evidence about the active ongoing disease. Early infections, which otherwise cannot be diagnosed through serology, can easily be analyzed through PCR. This test (real-time PCR) allows the quantitative evaluation of rickettsial pathogens. Multiplex PCR has the ability to discover co-infections as well [110,126]. Moreover, to evaluate persistently infected (subclinical form) carrier animals and for evaluating the efficacy of different chemotherapeutic trials, PCR is mandatory [127]. In past, genus-specific disulfide bond formation protein (dsb) gene were targeted for diagnosis [128]. Regarding *E. canis,* many molecular assays have been established targeting species-specific genes like p30 and 16S rRNA genes. p30 is considered more specific than 16SrRNA gene based on nested PCR. However, (IFAT) is considered the gold standard test by OIE for the diagnosis of *Ehrlichia* [129].

## 9. Treatment

Among antibiotics, tetracyclines were the first group to be used successfully against CME [16]. Doxycycline (semi-synthetic tetracycline) is approved experimentally both in vitro and in naturally infected dogs as the first-line treatment of choice. Ideal dose rate and duration of doxycycline is 10 mg/kg once (*per os*) daily or 5 mg/kg twice (orally) daily minimum for four weeks. This protocol warrants a satisfactory response in most of the cases [97,130]. In some studies, it has been approved that if we reduced the duration of treatment with doxycycline, complete elimination of *E. canis* became impossible and dogs played their role as carriers without displaying any clinical manifestations [131,132]. Thus, a complete four-week treatment protocol is recommended. As opposed to achieving clinical improvement, systematic removal of infection is difficult to accomplish, especially in naturally diseased dogs. Therefore, rather than post-treatment seronegativity, PCR negative results should be the target of clinicians [133]. 

In contrast to other tetracyclines, doxycycline is considered safe and does not cause discoloration of enamel. Vomiting is the second most common side effect of these antibiotics; this can be minimized by dividing the dose into two halves or by giving it after feeding. Prolonged use of doxycycline can damage hepatocytes, so if the liver enzymes increase, treatment must be paused [124,134]. The efficacy of minocycline has recently been evaluated for the chemotherapy of CME and it has shown a similar efficacy to doxycycline by eliminating the infection from all five dogs [135]. However, due to the very small sample size, it is suggested that more comprehensive studies should be conducted to declare minocycline as the first line of drug against CME [136]. Historically, chloramphenicol has also been used for treating ehrlichiosis in young dogs (less than 1 year of age). It is not recommended to administer chloramphenicol when doxycycline is accessible. In the past, imidocarb dipropionate was also considered effective against canine ehrlichiosis but more recent studies have revealed that it was only effective during co-infections and not against *E. canis* [137,138]. Rifampicin has been potentially useful in experimentally induced in vitro studies [139]. Further studies concluded that rifampicin can only decrease the intensity of clinical manifestations but could not eliminate the pathogen thoroughly from cells [140].

## 10. Post-Treatment Monitoring

Post-treatment supervision is chiefly vital in ehrlichiosis. Dissimilar to the myelosuppressive chronic form where treatment is largely ineffective, fast recovery (within 24–48 h) in the acute form is observed after the first dose, whereas blood cell abnormalities can be resolved within one to three weeks [131,141]. Another essential fact which we should keep in mind is that retrieval of hematological and clinical irregularities may surpass the exclusion of *E. canis*, so, the complete four-week treatment regime should be followed even when animal apparently looks normal. Recurrence of thrombocytopenia after the termination of doxycycline therapy shows failure of chemotherapy [142]. Constant presence of hyperglobulinemia even after 6 to 9 months of treatment stipulates coexisting infections or unsuccessful treatment. Quantitative serological tests cannot be used as a monitoring tool after treatment because values of IgG antibodies fluctuate randomly and remain there for months or years [143]. Therefore, from a clinical perspective, molecular detection (PCR) of bacteria is a reliable tool to declare clearance [144].

## 11. Prevention and Control

Even after the complete recovery from natural infection, dogs do not develop life-long immunity and there are still chances of re-infection [131]. That is why vector control by spraying suitable acaricides at regular intervals, by careful removal of ticks manually, or by monitoring of environmental factors related to tick growth, are fundamental control procedures in handling ehrlichiosis [145]. Recently, a study mentioned that transmission of *E. canis* may start after three to eight hours of tick infestation [144]. Commercially available ectoparasitic drugs such as pyrethroids (tetramethrin, permethrin, flumethrin, deltamethrin), phenylpyrazoles (fipronil, pyriprol), isoxazolines (sarolaner, fluralaner, afoxolaner), and amitraz have shown good efficacy regarding the killing of ticks involved in disease spread. However, owners must be educated about the development of resistance against these drugs and should encourage alternative use of different acaricides [144,145]. In regions where the disease is endemic, prophylactic use of doxycycline especially during summer and spring (tick season) can lower the risk of infection [145]; however, the probability of resistance increases with this practice. Dogs traveling from endemic areas must be screened for CME before entering.

## 12. Future Perspectives and Recommendations

The above discussion shows that ehrlichiosis in dogs is a vector-borne rickettsial ailment of zoonotic importance spreading globally. It is necessary to pay attention to canine ehrlichiosis and further research should focus on the following areas:(1)Molecular epidemiological surveys to investigate other tick species responsible for disease spread in different regions of the world for devising better control measures.(2)The role of other wild carnivores should be explored as a reservoir for this bacterium.(3)Further studies on finding the exact mechanism of immune envision.(4)The zoonotic potential of some ehrlichial species such as *E. canis* is not fully discovered yet, so health-based studies should be conducted.(5)Further genome-based studies to find pathogenic pathways/proteins for vaccine development.(6)Up to now, only one antibiotic class is mainly effective, so studies on finding new antibiotics are important.(7)In subtropical and tropical areas with an abundance of vectors, infection by two or more vector-borne pathogens is typical. Co-infections may accelerate the progression of a disease, changing the clinical signs and symptoms that are usually connected to a single infection. If the practitioner neglects to suspect, record, and treat each concurrent infection, these factors confound diagnosis, treatment, and can negatively affect prognosis. Canine ehrlichiosis has often been diagnosed with anaplasmosis or babesiosis in dogs. Thus, this co-infection halts the diagnosis of the exact etiological agent participating in a certain pathology, so pathophysiology and clinical presentation during co-infections need to be explored.

## 13. Conclusions

This review summarizes the current literature available on prevalence, geographical distribution, pathogenesis, epidemiology, and treatment as well as diagnostic methods of ehrlichiosis in dogs with special emphasis on east and south Asia. Data accessibility and availability regarding canine ehrlichiosis varies greatly between different countries of south and east Asia. Some countries, such as China and India, have profound availability of data about *Ehrlichia*; however, in Sri Lanka, Bhutan, Afghanistan, and North Korea, data are unavailable. Much of the literature reviewed about east Asian countries is very old, indicating the need for new research. In short, it is concluded that further research is needed with special emphasis on novel diagnostic tools, pathogenesis, epidemiology, disease transmission, and zoonoses regarding ehrlichiosis to explore a potential era for future studies.

## Figures and Tables

**Figure 1 vetsci-10-00021-f001:**
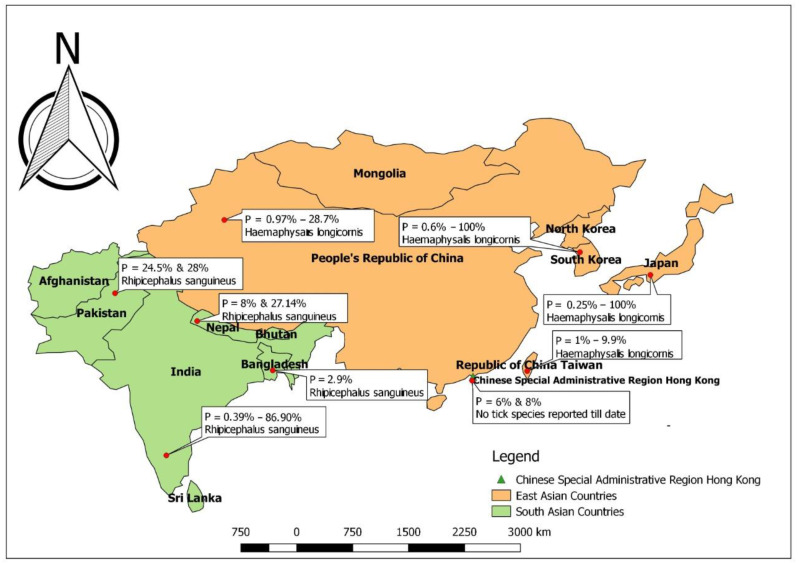
Prevalence percentage and main vector responsible for ehrlichiosis in dogs reported in different countries of east and south Asia.

**Table 1 vetsci-10-00021-t001:** Prevalence studies on ehrlichiosis in dogs conducted in south and east Asian countries.

Country/Region	Sample Source	Causative Agent/Species	Diagnostic Method	Prevalence % (Positive/Total Number of Samples)	Reference
South Asia Pakistan	Blood	*Ehrlichia canis*	PCR	28 (42/151)	[26]
Pakistan	Blood	*Ehrlichia canis*	PCR	24.5 (12/49)	[27]
India	Blood	*Ehrlichia canis*	PCR	8 (12/150)	[28]
India	Blood	*Ehrlichia canis*	PCR	8.40 (70/833)	[29]
India	Ticks Blood	*Ehrlichia canis* *Ehrlichia canis*	PCR PCR	16.1 16.9	[30]
India	Blood	*Ehrlichia canis*	PCR	30 (18/60)	[22]
India	Blood	*Ehrlichia canis*	PCR	41.59 (89/214)	[31]
India	Blood Serum	*Ehrlichia sp.* *Ehrlichia canis*	Microscopy ELISA	14.28 (12/84) 86.90 (73/84)	[24]
India	Blood Blood	*Ehrlichia sp.* *Ehrlichia canis*	Microscopy PCR	19.38 (19/98) 50 (49/98)	[32]
India	Blood	*Ehrlichia canis*	PCR	0.39 (3/778)	[33]
India	Serum	*Ehrlichia canis*	ELISA	48.33 (29/60)	[34]
India	Blood	*Ehrlichia canis*	PCR	20.6	[14]
India	Serum Blood	*Ehrlichia canis* *Ehrlichia canis*	ELISA PCR	57.5 (293/510) 8.8 (45/510)	[35]
India	Blood Serum Blood	*Ehrlichia canis* *Ehrlichia canis* *Ehrlichia canis*	Microscopy ELISA PCR	1.33 (3/225) 19.11 (43/225) 5.78 (13/225)	[36]
India	Serum	*Ehrlichia canis*	ELISA	19 (9/48)	[37]
Nepal	Blood	*Ehrlichia canis*	PCR	27.14 (19/70)	[38]
Nepal	Blood	*Ehrlichia spp.*	Microscopy	8 (4/50)	[39]
Bangladesh	Blood	*Anaplasma/* *Ehrlichia spp.*	PCR	2.9 (3/68)	[40]
East Asia People’s Republic of China (PRC)	Serum	*Ehrlichia canis*	ELISA	1.29 (4/309)	[41]
PRC	Blood Blood	*Ehrlichia canis* *Ehrlichia canis*	PCR PCR	12.12 (36/297) 15.23 (108/709)	[42]
PRC	Blood Ticks	*Ehrlichia canis* *Ehrlichia canis*	PCR PCR	1.4 (15/1114) 4.1 (6/146)	[43]
PRC	Ticks Serum	*Ehrlichia canis* *Ehrlichia canis* *Ehrlichia canis* *Ehrlichia canis*	PCR PCR PCR ELISA	11.03 (50/453) 3.29 (3/91) 13.69 (10/73) 1.33 (7/526)	[44]
PRC	Serum	*Ehrlichia canis*	SNAP test (EISA)	2.17 (13/600)	[45]
PRC	Ticks	*Ehrlichia canis*	PCR	0.97 (3/308)	[46]
PRC	Ticks	*Ehrlichia spp. (E.canis and E.muris like)*	PCR	28.7 (24/849)	[47]
PRC	Blood	*Ehrlichia canis*	ELISA	1.9%	[48]
PRC	Blood	*Ehrlichia canis*	PCR	0.0 (0/162)	[49]
Japan	Serum	*Ehrlichia canis* *Ehrlichia chaffeensis* *Ehrlichia muris*	IFA IFA IFA	18 (27/150) 18.7 (28/150) 11.3 (17/150)	[50]
Japan	Ticks	*Ehrlichia platys*	PCR	9.4 (3/32)	[51]
Japan	Blood	*Ehrlichia platys*	PCR	1.5 (1/67) Yamaguchi 27.6 (24/87) Okinawa	[51]
Japan	Serum	*Ehrlichia muris*	IFA	3.6 (18/499)	[52]
Japan	Blood	*Ehrlichia spp./Anaplasma* *spp.*	PCR	1.5 (11/722)	[53]
Japan	Blood	*Ehrlichia canis*	PCR+ Electron Microscopy	100 (1/1)	[54]
Japan	Ticks	*Ehrlichia spp*	PCR	0.25 (3/1211)	[55]
Japan	Ticks	*Ehrlichia canis*	PCR	0.0 (0/1211)	[55]
South Korea	Serum Blood	*Ehrlichia canis* *Ehrlichia canis*	SNAP test (ELISA) PCR	4.7 (25/532) 0.0 (0/25)	[56]
South Korea	Serum	*Ehrlichia spp. (E. canis/E. ewingii)* *Ehrlichia chaffeensis*	ELISA/IFA ELISA/IFA	10.3 (228/2215) 2.3 (52/2215)	[57]
South Korea	Serum Serum Blood	*Ehrlichia canis* *Ehrlichia canis* *Ehrlichia canis*	ELISA IFA PCR	0.6 (1/182) 22.5 (41/182) 0.0 (0/182)	[58]
South Korea	Blood	*Ehrlichia canis*	ELISA	6.1 (14/229) Rural dogs 0.0 (0/692) Urban dogs	[59]
South Korea	Serum Blood	*Ehrlichia spp* *Ehrlichia chaffeensis*	ELISA PCR	7.56 (22/291) 3.09 (9/291)	[60]
South Korea	Blood	*Ehrlichia chaffeensis*	PCR	100 (2/2)	[5]
South Korea	Ticks	*Ehrlichia chaffeensis*	PCR	4.2% (26/611)	[57]
South Korea	Blood	*Ehrlichia canis*	(ELISA)	12.3 (29/236)	[61]
South Korea	Ticks	*Ehrlichia canis* *Ehrlichia chaffeensis*	PCR PCR	0.0 (0/1110) 0.0 (0/1110)	[25]
ROC Taiwan	Blood	*Ehrlichia canis*	ELISA	2 (2/101)	[62]
ROC Taiwan	Blood	*Ehrlichia canis*	ELISA	1.5	[48]
ROC Taiwan	Blood	*Ehrlichia canis*	ELISA	9.9 (34/344)	[63]
ROC Taiwan	Blood Blood Ticks	*Ehrlichia canis Ehrlichia canis Ehrlichia canis*	ELISA PCR PCR	11.4 (20/175) 8.6 (15/175) 1 (3/306)	[64]
Hong Kong SAR, China	Blood	*Ehrlichia canis*	PCR	8 (8/100) stray 6 (6/100) Pet	[65]

**Table 2 vetsci-10-00021-t002:** Reported tick species carrying ehrlichiosis in south and east Asian countries/regions.

Country/Region.	Tick Vector	Reference
India	*Rhipicephalus sanguineus,* *Rhipicephalus haemaphysaloides*	[30]
People’s Republic of China (PRC)	*Rhipicephalus sanguineus* *Haemaphysalis longicornis,* *Rhipicephalus haemaphysaloides* *Haemaphysalis bispinosa*	[43,44,46,47]
Japan	*Rhipicephalus sanguineus* *Haemaphysalis flava*	[79,80]
South Korea	*Haemaphysalis longicornis* *Ixodes nipponensis*	[25,57]
ROC Taiwan	*Haemaphysalis hystricis Haemaphysalis longicornis*	[64]

## Data Availability

Data is contained within the article.

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
