# Peer review of "Ehrlichiosis in Dogs: A Comprehensive Review about the Pathogen and Its Vectors with Emphasis on South and East Asian Countries"

_vetsci, 2022, doi:10.3390/vetsci10010021_

Round 1

Reviewer 1 Report

Review is very good but should be better suited. KEYWORDS MUST BE DIFFERENT FROM THE TITLE. Introduction should start talking about Erlichia organisms. You should reverse the first paragraph with the second. Table titles should be self-explanatory as well as figures.

Reviewer 2 Report

Comments on the reviewed manuscript “Ehrlichiosis in dogs: a comprehensive review about the pathogen and its vectors with emphasis on South and East Asian countries” submitted to the Veterinary Sciences

General comments

I appreciate the opportunity to review this interesting manuscript, which is an extensive review of Ehrlichiosis in South and East Asia. The manuscript is a review of etiopathogenic, epidemiological aspects, diagnostic methods, treatment, and prevention of Ehrlichiosis in dogs in South and East Asia. The subject is relevant and can be used by many types of readers, in various fields of health knowledge.

The review on Ehrlichiosis in animals is based on 145 articles and book chapters, covering an extensive period of scientific production, from 1978 to 2021. The manuscript is well written, but there are some words or sentences that need modifications, which I indicate in the text below.

Title: Delete the period in the title.

Line 63: What is the reference for this argument? “Canine ehrlichiosis gained worldwide importance in the 1970s when a huge population of military German Shepherd dogs died during the Vietnam war”.

Line 89: Do you mean South Korea?

Lines 107 to 112: It is confusing. Rephrased it.

Line 113: It is sex, not gender.

Line 144: “However, in a recent …”

Line 149: Instead of "flexible", did you mean "resistant"?

Line 153: “…are amino acids”

Lines 293 to 320: This is an excessively long paragraph. I suggest breaking it into two paragraphs.

Figure 1: Figure captions are regularly positioned after the figure.

Line 381: The word “data” is a duplicate.  

Line 383: Delete the word “work”.

Line 395: The list of references needs to be carefully corrected, because I found spelling errors, misspelled scientific names, letters in different formats, and other errors.

Reviewer 3 Report

The authors of the paper entitled "Ehrlichiosis in dogs: a comprehensive review about the pathogen and its vectors with emphasis on South and East Asian countries" aimed to illustrate a brief overview of canine ehrlichiosis, discussing the epidemiology of the disease in South Asia, the signs of illness (both the clinical and pathological one) and the transmission cycle of the pathogen. Moreover, they discussed the diagnosis and treatment of disease, showing also the future perspectives of the argument.

The novelty of the narrative review is good, the argument is treated with scientific rigor and the final recommendations are very pertinent. However, some missing parts could be added to increase the scientific interest in the article.

  1. I suggest adding a part on the zoonotic potential of Ehrlichia spp. bacteria, describing the consequences of infections in humans.
  2. Why the authors did not mention the anatomopathological findings of canine disease, including both gross and histopathological changes?
  3. In the section related to Diagnosis, the authors could illustrate the most recent diagnostic techniques such as metagenomics strategies, including Next Generation Sequencing.
  4. The theme of coinfections should be clearly stated. 

Round 2

Reviewer 3 Report

The authors made the required corrections, the manuscript is now ready for publication in its present form.